# Improving Generalization and Robustness in SNNs Through Signed Rate Encoding and Sparse Encoding Attacks

**Bhaskar Mukhoty**[1][*] **Hilal AlQuabeh**[1][*], **Bin Gu**[2][†]
[1]Mohamed bin Zayed University of Artificial Intelligence, Abu Dhabi, UAE
[2]School of Artificial Intelligence, Jilin University, China

## Abstract

Rate-encoded spiking neural networks (SNNs) are known to offer superior adversarial robustness compared to direct-encoded SNNs but have relatively poor generalization on clean input. While the latter offers good generalization on clean input it suffers poor adversarial robustness under standard training. A key reason for this difference is the input noise introduced by the rate encoding, which encodes a pixel intensity with $T$ independent Bernoulli samples. To improve the generalization of rate-encoded SNNs, we propose the *signed rate encoding* (sRATE) that allows mean centering of the input and helps reduce the randomness introduced by the encoding, resulting in improved clean accuracy. In contrast to rate encoding, where input restricted to $[0, 1]^d$ is encoded in $\{0, 1\}^{d \times T}$, the signed rate encoding allows input in $[-1, 1]^d$ to be encoded with spikes in $\{-1, 0, 1\}^{d \times T}$, where positive (negative) inputs are encoded with positive (negative) spikes. We further construct efficient *Sparse Encoding Attack* (SEA) on standard and signed rate encoded input, which performs $l_0$-norm restricted adversarial attack in the discrete encoding space. We prove the theoretical optimality of the attack under the first-order approximation of the loss and compare it empirically with the existing attacks on the input space. Adversarial training performed with SEA, under signed rate encoding, offers superior adversarial robustness to the existing attacks and itself. Experiments conducted on standard datasets show the effectiveness of sign rate encoding in improving accuracy across all settings including adversarial robustness. The code is available at https://github.com/BhaskarMukhoty/SignedRateEncoding

## 1 Introduction

Spiking neurons emulate the salient features of the biological neurons by maintaining a membrane potential that accumulates the weighted input spikes over time and sends output spikes whenever the potential exceeds a predetermined threshold, followed by a reset in the potential (Gerstner et al., 2014). A spike is represented by a single binary number, denoting its presence or absence. In contrast to a continuous activation such as Sigmoid or ReLU, communication of information through spikes simplifies the floating point vector inner product required between the weights and activations for computation of input to a neuron, to the accumulation of weights selected by the presence of spikes. Such computation can be efficiently implemented on neuromorphic hardware, obtaining significant energy saving, as demonstrated in specialized neuromorphic hardware, such as Intel's Loihi (Davies et al., 2018), IBM's TrueNorth (Akopyan et al., 2015) etc.

For static inputs such as images, SNN requires each pixel to be encoded by spikes of length $T$, where $T$ is known as network latency. The direct or constant encoding supplies the input in $\mathbb{R}^d$, as it is, for $T$ time steps and assumes that the first layer of the network works as an encoder, generating the spikes. However, whether neuromorphic hardware can support floating point inputs is debatable. Since there is no restriction in the input domain, direct encoding allows the normalization of the input, helping in better training.

---

[*]Email: {bhaskar.mukhoty, halquabeh}@gmail.com
[†]Corresponding Author: jsgubin@gmail.com

In rate encoding approach, the input $\mathbf{x} = (x_1, x_2, \cdots, x_d) \in [0, 1]^d$, has a domain restriction. To encode a pixel $x_i \in [0, 1]$ it generates $T$ independent Bernoulli spikes $(z_i^{(1)}, z_i^{(2)}, \cdots, z_i^{(T)}) \in \{0, 1\}^T$, treating $x_i$ as the bias, with $z_i^{(t)} \sim Ber(x_i)$, so that, the rate of spikes $\frac{1}{T} \sum_{t=1}^{T} z_i^{(t)}$ approximates $x_i$. Due to the introduction of noise, rate encoding yields lower standard accuracy compared to direct encoding, moreover, because of the input domain restriction, it does not allow normalization of the input. It is, however, empirically observed (Sharmin et al., 2020) and also theoretically proved (Mukhoty et al., 2024) that rate-encoded network offers robustness from adversarial attacks, even without adversarial training. In contrast, direct encoded networks offer no adversarial robustness without adversarial training. It makes rate-encoded networks relevant to applications at the risk of adversarial attacks, as adversarial training for large datasets can sometimes be computationally prohibitive.

For some background, an adversarial attack (Goodfellow et al., 2014) is a test time attack that finds minimal perturbation of the input in order to change the classifier output to an incorrect class. On the other hand, to introduce robustness against adversarial attacks, adversarial training (Madry et al., 2018) performs training with adversarially perturbed input but the correct label.

Intuitively, a key reason behind the robustness of rate-encoding is the obfuscation of the original input through Bernoulli noise, which dilutes the effect of small adversarial perturbations of the input space when encoding is performed. With increasing $T$, as the rate of spikes better approximates the input, the clean accuracy improves, but at the same time, it reduces the robustness to adversarial perturbations. A primary motivation for the present work is to reduce the randomness introduced by the encoding for a fixed $T$, which should improve the standard accuracy but may cause a slight compromise in robustness. We observe that the randomness introduced by rate encoding can be reduced through mean-centering of the data, as the variance of a Bernoulli random variable is highest when the bias is $0.5$, and any reduction in the bias should result in a reduction in randomness.

Further, since adversarial perturbation in the input space must pass through the encoding space, an attack on the encoding space should be more effective if compared under a transferable budget with input space attacks. From a model perspective, receiving adversarially perturbed encoded input also makes realistic sense, as the encoder need not be a part of the model. For example, Marchisio et al. (2021) demonstrates an adversarial attack on SNN by perturbing dynamic vision sensor (DVS) images belonging to a discrete space. In literature, encoding attacks have been also explored, for instance, Liang et al. (2021) presents a gradient-based adversarial attack on rate encoding, which does not incorporate a sparsity budget but introduces uncontrolled sparsity through a probabilistic sampling of a mask applied to the perturbations, where the mask probability is equal to the coordinate-wise magnitude of the gradient.

**Our Contributions:**

- Rate encoding requires inputs $\mathbf{x} \in [0, 1]^d$ for Bernoulli sampling, which cannot accommodate negative inputs that arise from mean-centering the data. We propose *signed rate encoding* that allows rate encoding to work with both positive and negative inputs in $[-1, 1]^d$ by generating signed spikes depending upon the sign of the input. We quantify the reduction in randomness following mean-centering and demonstrate the effectiveness of signed rate encoding in improving training and generalization performance (see Table 1).

- We introduce gradient-based $l_0$-norm adversarial attack in the encoding space, dubbed as Sparse Encoding Attack (SEA), that alters the encoded input under a given sparsity budget and jointly optimizes the encoding restriction requiring the perturbed encoded input belonging to $\{0, 1\}$ or $\{-1, 0, 1\}$ depending upon rate or signed rate encoding. Given the gradient information and sparsity budget, we prove that SEA maximizes the loss function under the first-order approximation of the loss. Further, to compare the $l_0$-norm restricted encoding attacks with $l_\infty$-norm restricted input space attacks, we compute expected $l_0$-budget given an $l_\infty$-budget (see section 4.3).

- We perform adversarial training of rate-encoded SNNs with SEA and other existing input space attacks such as FGSM and PGD. The empirical performance on standard static image datasets such as CIFAR-10, CIFAR-100, and SVHN reveals that SEA under signed rate encoding offers significantly higher robust accuracy compared to the existing methods (see Table 6).

## 2 BACKGROUND

**Neuronal Dynamics:** The neurons of the SNN following the Leaky Integrate and Fire (LIF) model (Gerstner et al., 2014) are governed by a differential equation similar to RC-parallel circuit, which after Euler forward discretization, given as:

$$u_i^{(l)}[t] = \beta(u_i^{(l)}[t-1] - s_i^{(l)}[t-1]u_{th}) + \sum_j w_{ij}s_j^{(l-1)}[t], \tag{1}$$

$$s_i^{(l)}[t] = H(u_i^{(l)}[t] - u_{th}) = \begin{cases} 1 & \text{if } u_i^{(l)}[t] > u_{th} \\ 0 & \text{otherwise,} \end{cases} \tag{2}$$

where, $u_i^{(l)}[t]$ denotes the membrane potential of the $i$-th neuron in $l$-th layer at the time step $t \in [T]$. The potential recursively depends upon its residual potential from the previous time step with a leaky factor $\beta$, where $0 < \beta \leq 1$, and spikes $s_j^{(l-1)}[t]$ received from layer $l-1$ weighted by $w_{ij}$. When membrane potential $u_i^{(l)}[t]$ exceeds a predetermined threshold $u_{th}$, it generates a binary spike $s_i^{(l)}[t]$ represented by the Heaviside $H$, followed by a reset in the membrane potential at the next time step. The total number of steps for which the network dynamics is executed is called the latency of the network, denoted by $T$. The inputs to the network are supplied for $T$ steps using some encoding technique discussed earlier, and the network output is computed by taking the temporal average of the final layer output over the latency period.

**Surrogate Training:** Let $h_\theta : \mathbb{R}^d \to \mathbb{P}(\mathcal{Y})$ denote a classifier parameterized by $\theta$, which given input $\mathbf{x}_i$ returns the probabilities of the class labels in $\mathcal{Y}$. Given training data $(\mathbf{x}_i, y_i) \in S$, the network minimizes the loss $\mathcal{L}$ over $\theta$, computed w.r.t the true labels $y_i$:

$$\min_\theta \frac{1}{|S|} \sum_{\mathbf{x}_i, y_i \in S} \mathcal{L}(h_\theta(\mathbf{x}_i), y_i) \tag{3}$$

We use direct training, where minimization is performed using gradient decent updates computed via back-propagation through time (BPTT) (Wu et al., 2018). However, due to the Heaviside function involved, which has zero gradient everywhere except at discontinuity, it is customary to use a differentiable surrogate function in the backward pass and Heaviside in the forward. The surrogate gradients can also be replaced with zeroth order derivative of the Heaviside (Mukhoty et al., 2023).

**Adversarial Examples:** Given classifier $h_\theta$, an additive adversarial perturbation $\boldsymbol{\delta}_\mathbf{x}$ intends to alter the predicted class for a given input $\mathbf{x}$:

$$\arg\max_y h_\theta(\mathbf{x})_y \neq \arg\max_y h_\theta(\mathbf{x} + \boldsymbol{\delta}_\mathbf{x})_y \tag{4}$$

Solving the above task while minimizing a norm on $\boldsymbol{\delta}_\mathbf{x}$ is NP-hard, owing to the non-convexity of the classifier $h_\theta$. Considering the white box attack scenario where the adversary can access the network architecture and parameters, one tries to find a perturbation $\boldsymbol{\delta}_\mathbf{x}$ using the loss as a proxy objective (Goodfellow et al., 2014),

$$\arg\max_{\|\boldsymbol{\delta}\| \leq \epsilon} \mathcal{L}(h_\theta(\mathbf{x} + \boldsymbol{\delta}), y) \tag{5}$$

Since the loss is non-convex, it is common to work with its first-order approximation, $\mathcal{L}(h_\theta(\mathbf{x} + \boldsymbol{\delta}), y) \simeq \mathcal{L}(h_\theta(\mathbf{x}), y) + \boldsymbol{\delta}^T \nabla_x \mathcal{L}(h_\theta(\mathbf{x}), y)$, in which case, the objective becomes:

$$\boldsymbol{\delta}_\mathbf{x} := \arg\max_{\|\boldsymbol{\delta}\| \leq \epsilon} \boldsymbol{\delta}^T \nabla_x \mathcal{L}(h_\theta(\mathbf{x}), y) \tag{6}$$

The Fast Sign Gradient Method (FGSM) (Goodfellow et al., 2014) solves the problem for $l_\infty$-norm restriction on $\boldsymbol{\delta}$. The problem can also attempt to be solved over multiple FGSM steps, each followed by projection operations $\Pi_\epsilon$ that enforce the norm restriction, known as Projected Gradient Descent (PGD) (Madry et al., 2018):

$$\boldsymbol{\delta}_{t+1} = \Pi_\epsilon(\boldsymbol{\delta}_t + \alpha \operatorname{sign}(\nabla_{\boldsymbol{\delta}} \mathcal{L}(h_\theta(\mathbf{x} + \boldsymbol{\delta}_t)), y)) \tag{7}$$

As the perturbed image must satisfy constraints of input space, e.g., $\mathbf{x} + \boldsymbol{\delta}_t \in [0, 1]^d$, it requires an additional projection. It is established that joint optimization of norm and the input space restriction

is superior to the sequential projections required to ensure them (Croce & Hein, 2021). Inspired by this, we perform a joint optimization these objectives in the proposed $l_0$-norm attack (see eqn. 14).

**Adversarial Training:** To train a classifier that performs reasonably well even under adversarial perturbations, it was proposed (Madry et al., 2018) that the classifier be trained with perturbed input $\mathbf{x} + \boldsymbol{\delta}_{\mathbf{x}}$ but with original label $y$.

$$\min_{\theta} \frac{1}{|S|} \sum_{\mathbf{x}, y \in S} \max_{\|\delta\| \leq \epsilon} \mathcal{L}(h_{\theta}(\mathbf{x} + \boldsymbol{\delta}), y) \tag{8}$$

Though computationally more expensive than standard training due to inner maximization required to find the perturbation, the objective forces classifier output to be invariant across the radius of perturbation, introducing a bias that often leads to a drop in clean accuracy, but offers better robust accuracy against the adversarial attack (Pang et al., 2022).

## 3 SIGNED RATE ENCODING

### 3.1 QUANTIFICATION OF RANDOMNESS

We start by quantifying the amount of randomness introduced by rate encoding (RATE). To this end, we would like to measure how far on expectation two binary encodings can be when measured under $l_0$-norm. Consider an image input $\mathbf{x} \in [0, 1]^d$, encoded twice independently at a particular time step, i.e., $\mathbf{z}, \hat{\mathbf{z}} \sim Ber(\mathbf{x})$, where, $\mathbf{z}, \hat{\mathbf{z}} \in \{0, 1\}^d$. So that we have,

$$\mathbb{E}_{\mathbf{z}, \hat{\mathbf{z}}}[\|\mathbf{z} - \hat{\mathbf{z}}\|_0] = \sum_{i=1}^{d} \mathbb{E}_{z_i, \hat{z}_i \sim Ber(x_i)}[|z_i - \hat{z}_i|] = 2 \sum_{i=1}^{d} x_i(1 - x_i) = 2 \langle \mathbf{x}, 1 - \mathbf{x} \rangle =: k_1(\mathbf{x}) \tag{9}$$

The randomness $k_1(\mathbf{x})$ is proportional to the sum of the pixel variances, e.g., if every pixel has an intensity of 0.5, two encodings at any particular time step $t$ will vary on $\frac{d}{2}$ coordinates on expectation. A natural question that follows is whether it is possible to reduce the randomness. We can reduce the variance of the pixels if we push them towards boundaries 0 or 1, as the variance is highest at 0.5. A standard way to do it is mean centering, making the pixels have 0 intensity on average. However, this will require negative intensities to be encoded, a case not supported by rate encoding.

### 3.2 SIGNED RATE ENCODING VIA SIGNED BERNOULLI

We propose *signed rate encoding* (SRATE) to allow negative as well as positive pixel intensities, $x_i \in [-1, 1]$, to be rate encoded. For this, whenever a pixel intensity is positive, we generate the spikes in $\{0, 1\}$ using Bernoulli sampling, while, if the intensity is negative, we generate spikes in $\{0, -1\}$, treating the magnitude of the intensity as the bias of a negative Bernoulli variable.

$$z_i = \mathrm{sBer}(x_i) := \begin{cases} \mathrm{Ber}(x_i) & \text{if } x_i \geq 0 \\ -\mathrm{Ber}(-x_i) & \text{if } x_i < 0 \end{cases} \tag{10}$$

so that, $z_i \in \{-1, 0, 1\}$, with,

$$\mathbb{P}(z_i = 1) = x_i \mathbb{I}[x_i \geq 0] = x_i^+, \qquad \mathbb{P}(z_i = -1) = -x_i \mathbb{I}[x_i < 0] = x_i^-$$
$$\mathbb{P}(z_i = 0) = 1 - x_i^+ - x_i^- = 1 - |x_i| \tag{11}$$

The notation, $x_i^+, x_i^-$, are respectively known as positive and negative part, both non-negative quantities, denoting the magnitude of the number with either sign, but zero otherwise. Eqn. 11 uses the identity $|x_i| = x_i^+ + x_i^-$ for a compact representation. We refer to the random variable defined as above (eqn. 10, 11) as *signed Bernoulli* (abbr. sBer) and the corresponding encoding as signed rate encoding (SRATE). Given an input $\mathbf{x} \in [-1, 1]^d$, it generates the encoding in $\{-1, 0, 1\}^{d \times T}$ by sampling independently from the sBer distribution for $T$ steps.

We quantify the randomness introduced by signed rate encoding similar to eqn. 9. Given $\mathbf{x} \in [-1, 1]^d$, we compute the expected $l_0$-norm distance of two independent signed Bernoulli encoding $\mathbf{z}, \hat{\mathbf{z}} \sim$

$\text{sBer}(\mathbf{x})$, at a fixed time step $t$ with, $\mathbf{z}, \hat{\mathbf{z}} \in \{-1, 0, 1\}^d$.

$$\mathbb{E}_{\mathbf{z}, \hat{\mathbf{z}} \sim \text{sBer}(\mathbf{x})}[\|\mathbf{z} - \hat{\mathbf{z}}\|_0] = \sum_{i=1}^{d} \mathbb{E}_{z_i, \hat{z}_i \sim \text{sBer}(x_i)}[\|z_i - \hat{z}_i\|_0] = \sum_{i=1}^{d} \mathbb{P}(z_i \neq \hat{z}_i) = \sum_{i=1}^{d}(1 - \mathbb{P}(z_i = \hat{z}_i))$$

$$= \sum_{i=1}^{d}(1 - \mathbb{P}(z_i = 1, \hat{z}_i = 1) - \mathbb{P}(z_i = -1, \hat{z}_i = -1) - \mathbb{P}(z_i = 0, \hat{z}_i = 0))$$

$$= \sum_{i=1}^{d}\left(1 - (x_i^+)^2 - (x_i^-)^2 - (1 - |x_i|)^2\right) = \sum_{i=1}^{d}\left(1 - |x_i|^2 - (1 - |x_i|)^2\right)$$

$$= 2\langle |\mathbf{x}|, \mathbf{1} - |\mathbf{x}| \rangle =: k_2(\mathbf{x}) \tag{12}$$

Note that eqn. 12 goes back gracefully to eqn. 9, with inputs restricted to $[0, 1]$.

Denoting the empirical mean as $\boldsymbol{\mu} = \frac{1}{n}\sum_{i=1}^{n}\mathbf{x}_i \in [0, 1]^d$, we compute the factor,

$$k_3(\mathbf{x}) := \frac{k_1(\mathbf{x})}{k_2(\mathbf{x} - \boldsymbol{\mu})} \tag{13}$$

which, represents the factor by which the randomness reduces following mean centering of a particular input $\mathbf{x}$.

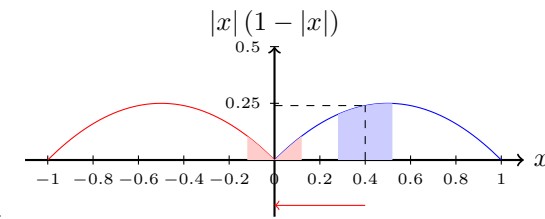

Figure 1: Signed Bernoulli reducing randomness after mean-centering of CIFAR-10 dataset.

Tab. 1 reports the quantities $k_1, k_2$ and $k_3$ after taking the average on the training set of multiple standard datasets, along with channel-wise average pixel intensities. The SVHN dataset shows the highest reduction in randomness, followed by ImageNet-100, CIFAR-10 and CIFAR-100. For CIFAR-100, although avg. $k_2(\mathbf{x} - \boldsymbol{\mu})$ is higher than avg. $k_1(\mathbf{x})$, we still obtain a reduction in randomness, as reported by avg. $k_3(\mathbf{x})$. Fig. 1 attempts to illustrate the possible reduction in randomness due to the mean centering of a dataset, with shaded areas highlighting the standard deviation of pixel intensities before and after the centering.

**Characterizing Randomness Reduction:** We note that the reduction in randomness offered by SRATE is a data-dependent result; such a reduction may not always happen. Tab. 5 shows that the rate of spikes in the input layer has a significant reduction in SRATE compared to RATE, which is possible when $|x_i - \mu| < x_i$ for most $x_i \in [0, 1]$, which also implies a reduction in randomness as the function $x(1 - x)$ is an increasing function in the region $[0, 0.5]$. Observe, $|x_i - \mu| < x_i \iff 0 < \mu < 2x_i$. As $\mu \geq 0$ is true for $x_i \in [0, 1]$, the condition fails only when $\mu > 2x_i$ or, $x_i > 0.5$, thus, $x_i \in [\frac{\mu}{2}, 0.5]$ guarantees randomness reduction. However, it is only a sufficient condition.

| | CIFAR-10 | CIFAR-100 | SVHN | ImageNet-100 |
|---|---|---|---|---|
| $d$ | $32 \times 32 \times 3$ | $32 \times 32 \times 3$ | $32 \times 32 \times 3$ | $224 \times 224 \times 3$ |
| pixel | (0.425, 0.415, 0.384) | (0.439, 0.418, 0.378) | (0.378, 0.384, 0.410) | (0.458, 0.450, 0.389) |
| $k_1(\mathbf{x})$ | 994.18 | 953.09 | 1106.59 | 54701.22 |
| $k_2(\mathbf{x} - \mu)$ | 942.39 | 964.14 | 830.33 | 43998.53 |
| $k_3(\mathbf{x})$ | 1.11 | 1.05 | 1.44 | 1.366 |

Table 1: Dataset statistics and their randomness reduction factor ($k_3$) due to signed rate encoding averaged over training data points.

## 4 SEA: SPARSE ENCODING ATTACKS

### 4.1 ADVERSARIAL ATTACK ON RATE ENCODING

It is natural for an SNN classifier to receive encoded inputs, thus creating a possibility of an adversarial attack on the encoded input. Let us consider the classifier $h_\theta : \{0, 1\}^{d \times T} \to \mathbb{P}(\mathcal{Y})$, that receives rate-encoded input $\mathbf{z} = [\mathbf{z}^{(1)}, \mathbf{z}^{(2)}, \cdots, \mathbf{z}^{(T)}]$, with $\mathbf{z}^{(t)} \in \{0, 1\}^d$ and produces probabilities for

| sign($g_i$) | -1 | -1 | 1 | 1 | 0 | 0 |
|---|---|---|---|---|---|---|
| $z_i$ | 0 | 1 | 0 | 1 | 0 | 1 |
| $\hat{\delta}_i$ | 0 | -1 | 1 | 0 | 0 | 0 |

Table 2: $\hat{\delta}_i + z_i \in \{0, 1\}$

| sign($g_i$) | -1 | -1 | -1 | 1 | 1 | 1 | 0 | 0 | 0 |
|---|---|---|---|---|---|---|---|---|---|
| $z_i$ | -1 | 0 | 1 | -1 | 0 | 1 | -1 | 0 | 1 |
| $\hat{\delta}_i$ | 0 | -1 | -2 | 2 | 1 | 0 | 0 | 0 | 0 |

Table 3: $\hat{\delta}_i + z_i \in \{-1, 0, 1\}$

individual classes. To construct an untargeted attack on the encoding $\mathbf{z} \in \{0,1\}^{d \times T}$, we may want to restrict the changes to at most $k$ coordinates in a frame $\mathbf{z}^{(t)} \in \{0,1\}^d$. Thus, we would like to solve the following optimization problem, following the first-order approximation of the loss function as described in eqn. 6.

$$\boldsymbol{\delta}^* := \underset{\boldsymbol{\delta}: \forall t \, \|\boldsymbol{\delta}^{(t)}\|_0 \leq k, \mathbf{z}^{(t)} + \boldsymbol{\delta}^{(t)} \in \{0,1\}^d}{\arg\max} \langle \boldsymbol{\delta}, \nabla_{\mathbf{z}} \mathcal{L}(h_\theta(\mathbf{z} + \boldsymbol{\delta}), y) \rangle \tag{14}$$

where $\boldsymbol{\delta} = [\boldsymbol{\delta}^{(1)}, \boldsymbol{\delta}^{(2)}, \ldots, \boldsymbol{\delta}^{(T)}]$, and the restriction $\mathbf{z}^{(t)} + \boldsymbol{\delta}^{(t)} \in \{0,1\}^d$ implies that $\boldsymbol{\delta}^{(t)} \in \{-1,0,1\}^d$. Since the optimization problem is over a finite and non-empty feasible set, it must attain a global maximum. However, the argument on which it attains the maxima may not be unique, and we would like to find one of them and call it $\boldsymbol{\delta}^*$. But first, let us solve the problem without the sparsity constraint.

$$\hat{\boldsymbol{\delta}} := \underset{\boldsymbol{\delta}: \forall t \, \mathbf{z}^{(t)} + \boldsymbol{\delta}^{(t)} \in \{0,1\}^d}{\arg\max} \langle \boldsymbol{\delta}, \nabla_{\mathbf{z}} \mathcal{L}(h_\theta(\mathbf{z} + \boldsymbol{\delta}), y) \rangle \tag{15}$$

As the objective is linear in $\boldsymbol{\delta}$, the solution of the problem can be given coordinate-wise. With the renaming $\mathbf{g} := \nabla_{\mathbf{z}} \mathcal{L} \in \mathbb{R}^{d \times T}$, the solution is given as:

$$\hat{\delta}_i^{(t)} = \begin{cases} 1 & g_i^{(t)} > 0, z_i^{(t)} = 0 \\ -1 & g_i^{(t)} < 0, z_i^{(t)} = 1 \\ 0 & \text{otherwise} \end{cases}$$
$$= \text{sign}(g_i^{(t)})(\mathbb{I}[g_i^{(t)} \geq 0](1 - z_i^{(t)}) + (1 - \mathbb{I}[g_i^{(t)} \geq 0])z_i^{(t)}) \tag{16}$$

Table 2, elaborates the solution by observing that once we fix $z_i^{(t)}$ and $g_i^{(t)}$, the choice of $\hat{\delta}_i^{(t)}$ that maximizes the quantity $\hat{\delta}_i^{(t)} g_i^{(t)}$ while respecting the constraint $z_i^{(t)} + \delta_i^{(t)} \in \{0, 1\}$ is straightforward.

**Lemma 1.** *The solution $\hat{\boldsymbol{\delta}}$ defined as per eqn. 16 is an optimal solution to the optimization problem in eqn. 15.*

We give the proof in the supplementary section A.1.

With $\hat{\boldsymbol{\delta}}$ present in hand, it is not difficult to find $\boldsymbol{\delta}^*$. To maximize the inner product $\langle \hat{\boldsymbol{\delta}}, \mathbf{g} \rangle$ under a sparsity budget, we would like to choose the top k no-zero coordinates of $\hat{\boldsymbol{\delta}}^{(t)}$, according to the magnitude $\hat{\delta}_i^{(t)} g_i^{(t)}$. We define a permutation $\pi$ specific to the time step $t$ such that, $i < j$ implies $\hat{\delta}_{\pi_i}^{(t)} g_{\pi_i}^{(t)} \geq \hat{\delta}_{\pi_j}^{(t)} g_{\pi_j}^{(t)}$, for $i, j \in [1, 2, \cdots, d]$. $\boldsymbol{\delta}^{*(t)}$ keeps the top $k$ coordinates of $\hat{\boldsymbol{\delta}}^{(t)}$ with respect to $\pi$ and sets the rest to zero:

$$\delta_{\pi_i}^{*(t)} = \begin{cases} \hat{\delta}_{\pi_i}^{(t)} & \text{if} \quad i \leq k \\ 0 & \text{otherwise} \end{cases} \tag{17}$$

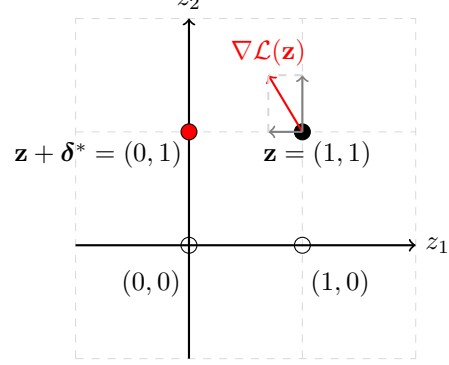

Figure 2: Sparse Encoding Attack on rate encoding for a given time-step, with d=2 and k=1

**Lemma 2.** $\boldsymbol{\delta}^*$ *obtained as per eqn. 17 is a maximizer of the optimization problem of eqn. 14.*

**Time Complexity:** To implement the operation in eqn. 17 one needs to find the indices of top-k elements of an array which holds the elements

$[\hat{\delta}_1^{(t)} g_1^{(t)}, \cdots, \hat{\delta}_d^{(t)} g_d^{(t)}]$. It can be implemented with a partitioning algorithm with the time complexity of $O(d + k)$, making the attack as efficient as other gradient-based attacks.

Highlighting the sparse nature of the attack on the encoding, we name it *Sparse Encoding Attack* or SEA. Fig. 2 gives a visual example of the attack, where we intend to perform a 1-sparse perturbation at the point $\mathbf{z} = (1, 1)$ on which the gradient is assumed to be $(-0.5, 1)$. Although the gradient has the largest component in the direction of $z_2$, any positive progress in that direction is not possible due to the violation of the feasibility condition. But, in the direction of $z_1$, a change is possible, and SEA gives us $\boldsymbol{\delta}^* = (-1, 0)$. Next, we construct the same attack adapted to signed rate encoding.

### 4.2 SPARSE ENCODING ATTACK ON SIGNED RATE ENCODING

To construct *sparse encoding attack* on signed rate encoding, we change the encoding space to $\{-1, 0, 1\}^d$, fixing a time step $t$, so that eqn. 14 is reformulated as:

$$\boldsymbol{\delta}^* := \underset{\boldsymbol{\delta}: \forall t \, \|\boldsymbol{\delta}^{(t)}\|_0 \leq k, \mathbf{z}^{(t)} + \boldsymbol{\delta}^{(t)} \in \{-1,0,1\}^d}{\arg \max} \langle \boldsymbol{\delta}, \nabla_{\mathbf{z}} \mathcal{L}(h_\theta(\mathbf{z} + \boldsymbol{\delta}), y) \rangle \tag{18}$$

The feasibility constraint ensures that, $\boldsymbol{\delta}^{(t)} \in \{-2, -1, 0, 1, 2\}^d$. Similar to rate encoding, we first construct a coordinate-wise solution to the problem without the sparsity restriction, similar to eqn. 16 and given by Tab. 3. It is can be compactly expressed as:

$$\hat{\delta}_i^{(t)} = \text{sign}(g_i^{(t)}) - z_i^{(t)} \left| \text{sign}(g_i^t) \right| \tag{19}$$

Given $\hat{\boldsymbol{\delta}}$, $\boldsymbol{\delta}^*$ can be obtained following the definition of eqn. 17. The optimality of the solution can be proven by a similar argument as presented in Theorem 2.

### 4.3 COMPARISON OF SEA WITH INPUT SPACE ATTACKS

To compare input space attacks with encoding attacks, given any adversarial perturbation $\boldsymbol{\delta}_{\mathbf{x}}$ in input space, we need to know the change in sparsity it may cause in the encoding space. To quantify this for rate encoding, we consider a corrupted input vector $\hat{\mathbf{x}} = \mathbf{x} + \boldsymbol{\delta}_{\mathbf{x}}$ with $\|\boldsymbol{\delta}_{\mathbf{x}}\| \leq \epsilon$, $\hat{\mathbf{x}} \in [0, 1]^d$, and find $\mathbb{E}[\|\mathbf{z} - \hat{\mathbf{z}}\|_0]$, with the assumption, $\mathbf{z} \sim Ber(\mathbf{x})$ and $\hat{\mathbf{z}} \sim Ber(\hat{\mathbf{x}})$.

$$\hat{k}_1(\mathbf{x}, \delta_{\mathbf{x}}) = \mathbb{E}_{\mathbf{z}, \hat{\mathbf{z}}}[\|\mathbf{z} - \hat{\mathbf{z}}\|_0] = \sum_{i=1}^d \mathbb{E}[|z_i - \hat{z}_i|] = \sum_{i=1}^d (1 - x_i)\hat{x}_i + x_i(1 - \hat{x}_i)$$

$$= \sum_{i=1}^d (1 - x_i)(x_i + \delta_i) + x_i(1 - x_i - \delta_i) = 2\langle \mathbf{x}, 1 - \mathbf{x} \rangle + \langle \boldsymbol{\delta}_{\mathbf{x}}, 1 - 2\mathbf{x} \rangle \tag{20}$$

The expression for $\hat{k}_1(\mathbf{x}, \delta_{\mathbf{x}})$ reduces to $k_1(\mathbf{x})$ (see eqn. 9) as we set $\delta_{\mathbf{x}} = 0$. Thus, the quantity $k := \left| \hat{k}_1(\mathbf{x}, \delta_{\mathbf{x}}) - k_1(\mathbf{x}) \right| = |\langle \boldsymbol{\delta}_{\mathbf{x}}, 1 - 2\mathbf{x} \rangle|$ gives the expected sparsity change on the encoding of $\mathbf{x}$ due to input perturbation $\delta_{\mathbf{x}}$. Fig. 4(d) plots this quantity averaged over the training dataset, where $\delta_{\mathbf{x}}$ are found for each $\mathbf{x}$ by FGSM/PGD attack, under different values of $\epsilon$ shown in the x-axis. It can observed from the figure, that for an $l_\infty$ attack of radius $\epsilon = \frac{8}{255}$, the expected change in sparsity is $k = 10$. Thus, it is fair to compare a $\epsilon = \frac{8}{255}$ FGSM/PGD attacks with SEA at $k = 10$, under the rate encoding. See Tab. 6 for such a comparison.

For computation of the sparsity budget for SRATE, we obtain:

$$\hat{k}_2(\mathbf{x}, \delta_{\mathbf{x}}) = \mathbb{E}_{\mathbf{z} \sim sBer(\mathbf{x}), \hat{\mathbf{z}} \sim sBer(\hat{\mathbf{x}})}[\|\mathbf{z} - \hat{\mathbf{z}}\|_0]$$
$$= \|\mathbf{x}\|_1 + \langle 1 - |\mathbf{x}|, |\mathbf{x} + \boldsymbol{\delta}| \rangle - \langle \mathbf{x}^+, (\mathbf{x} + \boldsymbol{\delta})^+ \rangle - \langle \mathbf{x}^-, (\mathbf{x} + \boldsymbol{\delta})^- \rangle \tag{21}$$

where, the derivation is given in appendix section A.2. $\hat{k}_2(\mathbf{x}, \delta_{\mathbf{x}})$ also reduces to $k_2(\mathbf{x})$ with $\delta_{\mathbf{x}} = 0$. Thus, giving effective sparsity of the SEA attack, as $k = \left| \hat{k}_2(\mathbf{x}, \delta_{\mathbf{x}}) - k_2(\mathbf{x}) \right|$, as plotted in Fig. 4(d).

## 5 EXPERIMENTS

Our experiments attempt to cover two aspects: first, the improvement in generalization by signed rate encoding over standard rate encoding, and secondly, the effectiveness of *sparse encoding attack* and

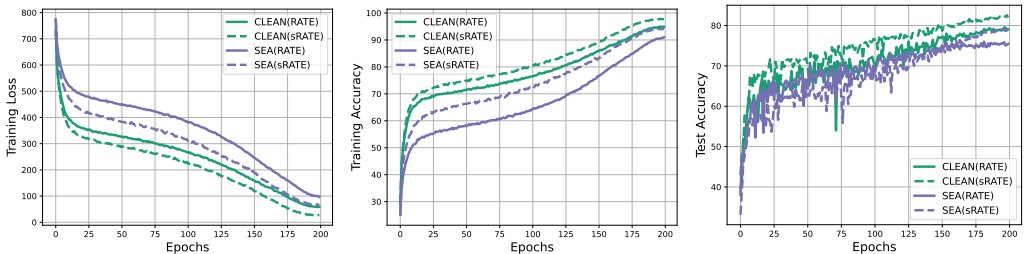

Figure 3: We compare the performance of rate encoding (RATE) and signed rate encoding (SRATE) on CIFAR-10. It can be observed that a significant improvement in training is offered by SRATE, and it is reflected consistently across the training loss, training accuracy and test accuracy. A similar observation can be made on adversarial training with SEA.

it's adversarial training. We use standard datasets such as CIFAR-10/100 (Krizhevsky et al., 2009), SVHN (Netzer et al., 2011) and Imagenet-100 (Deng et al., 2009) for comparing the algorithms. Following recent work we use the standard VGGSNN (Deng et al., 2022) architecture based on VGG-11 having the configuration (64C3-128C3-AP2-256C3-256C3-AP2-512C3-512C3-AP2-512C3-512C3-AP2-FC) for all datasets, except for Imagenet-100 where we use SEW-Resnet-34 (Fang et al., 2021).

**Competitors:** We compare with standard $l_\infty$-norm based input space attacks such as FGSM (Goodfellow et al., 2014) and PGD (Madry et al., 2018). Implementation of the input space attacks requires back-propagation of gradient through the stochasticity of $Ber/\mathrm{sBer}$. We use the straight-through estimator (Bengio et al., 2013), as demonstrated for input space attacks with rate encoding (Mukhoty et al., 2024).

**Training:** Complementary to standard training on clean data (CLEAN) and additive Gaussian Noise (GN), we perform adversarial training with FGSM, PGD and SEA attacks. The training and test attack radii for each attack are supplied in Tab.10, along with complete details on the training hyper-parameters. Since both attack and adversarially trained models have the same name, we denote the attacks with small cases, while the models are highlighted in the capital within the figures and Tab.6. We use the Back Propagation Through Time (BPTT) (Wu et al., 2018) algorithm for finding gradients for adversarial attack and model parameters. A comparison of BPTT with BPTR (Back Propagation Through Rate) Ding et al. (2022) supplied in Tab. 11, shows the BPTT attack to be superior, except in two cases, where BPTR performs better in the CIFAR-100 dataset.

## 5.1 SIGNED RATE ENCODING VS RATE ENCODING:

We compare the training loss, training accuracy and test accuracy at the end of each training epoch as offered by SRATE and RATE in three datasets. Fig.3 reveals that the loss converges much faster in SRATE compared to RATE, in CIFAR-10. Also evident from the figure, the advantage in training is reflected in test accuracies, where SRATE is found to offer higher accuracy than RATE, consistently after each epoch. A similar trend can be found for adversarial training, as depicted under the SEA attack. Figures for CIFAR-100 and SVHN datasets are supplied in the appendix (see Fig. 5), which show approximately similar results.

| T | RATE | SRATE | Diff. |
|----|-------|-------|-------|
| 4 | 78.79 | 82.05 | 3.26 |
| 6 | 82.22 | 84.41 | 2.19 |
| 8 | 83.41 | 85.41 | 2.00 |
| 10 | 84.21 | 86.34 | 2.13 |

Table 4: Comparison of clean test accuracy on different latencies for CIFAR-10, shows improvement offered by signed rate encoding over rate encoding.

To see the exact improvement in standard accuracies on different datasets, we need to compare the columns represented by CLEAN in Tab 6 and rows represented by clean. Such comparison shows that SRATE offers 3.26%, 4.18% and 3.76% higher standard / clean accuracy for CIFAR-10, 100 and SVHN than RATE. Improvement in generalization is also reported for ImageNet-100, where test accuracy is improved by 1%, as reported in Tab. 7.

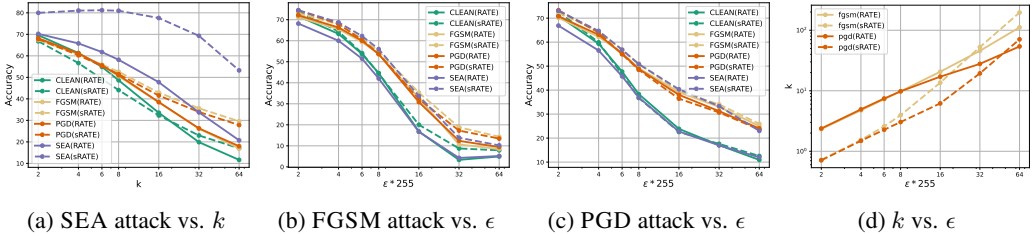

| (a) SEA attack vs. $k$ | (b) FGSM attack vs. $\epsilon$ | (c) PGD attack vs. $\epsilon$ | (d) $k$ vs. $\epsilon$ |
|---|---|---|---|

Figure 4: We fix the adv. trained models (highlighted by the legends) of CIFAR-10 dataset from Tab. 6 and perform sensitivity study on attack strength by varying (a) $k$ in SEA, (b) $\epsilon$ in FGSM and (c) $\epsilon$ in PGD. Observe, (i) SRATE models offering higher robustness than corresponding RATE model and (ii) adv. training with SEA and SRATE offering much higher robustness under SEA attack and comparable robustness against other attacks. Please refer to Tab. 13 for exact numbers in the plots.

The advantages of SRATE over RATE, is also reflected when compared under different training latencies, as shown in Tab. 4 reporting clean test accuracy for CIFAR-10 dataset.

**Comparison of Sparsity:** We compute the layer-wise spiking rate of the neurons, where the number of spikes in a neuron is normalized by latency T and averaged over the neurons in the layer. Tab. 5 compares the statistics for RATE and SRATE at inference time on different datasets with their respective networks (SEWResNet for Imagenet-100, VGGSNN for others) for CLEAN models. We can observe that spikes in the input layer have reduced significantly in SRATE due to mean centering. Further, the spike rate averaged over all neurons in the network remains comparable or slightly higher for SRATE.

| Dataset | Encoding | Input | Avg. |
|---|---|---|---|
| CIFAR-10 | RATE | 0.48 | 0.053 |
| | SRATE | 0.22 | 0.061 |
| SVHN | RATE | 0.46 | 0.058 |
| | SRATE | 0.20 | 0.062 |
| CIFAR-100 | RATE | 0.48 | 0.082 |
| | SRATE | 0.23 | 0.085 |
| ImageNet-100 | RATE | 0.43 | 0.083 |
| | SRATE | 0.22 | 0.084 |

Table 5: Rate of spikes in the input layer shows a significant reduction in SRATE, while it slightly increases the average spiking activity in the network. Detailed results are provided in Tab. 9

## 5.2 EFFECTIVENESS OF SPARSE ENCODING ATTACKS

Tab. 6 shows application of SEA on RATE and SRATE, both as an attack and with adversarial training. The sparsity budget $k = 10$ for SEA in RATE is found to be comparable with $l_\infty$-attack radius of $\frac{8}{255}$, as shown in Fig.4(d) and discussed in section 4.3. For each of the $T$ frames, SEA is allowed to change at most $k$ pixels out of $d$, for a reference, in CIFAR-10, with $d = 32 \times 32 \times 3 = 3072$, where $k = 10$ amount to changing only $0.326\%$ of pixels per frame. Observe that the SEA has a comparable performance to FGSM attack, even with $k = 10$, and significantly beats all attacks with the budget $k = 20$.

The columns of Tab. 6 compare different adversarial training algorithms, under the mentioned attacks. It can be observed that adv. training with SEA under SRATE, offers higher robust accuracy compared to other models and across the attacks, as highlighted in the last column. To explain, why SEA under RATE, is not as good, one may refer to Fig. 4 (d) and Tab. 14, which shows that the equivalent sparsity for $\epsilon = \frac{8}{255}$ is 10 for RATE, while it is 4 in SRATE. We perform adversarial training with SEA at $k = 5$ for both RATE and SRATE, possibly causing SEA model under RATE to perform worse under attacks of magnitude equivalent to 10.

We consider the adversarially trained models of CIFAR-10 from Tab. 6 and perform sensitivity studies on attack strength by varying $k$ in SEA and $\epsilon$ in FGSM and PGD. We can observe

| T=4 | ImageNet-100 | |
|---|---|---|
| Attack | RATE | SRATE |
| clean | 71.22 | 72.22 |
| gn | 71.38 | 72.06 |
| fgsm | 22.54 | 21.76 |
| pgd | 11.04 | 11.62 |
| sea, k=10 | 67.06 | 65.4 |
| sea, k=20 | 64.64 | 61.44 |

Table 7: Comparison of signed rate encoding with rate encoding trained on clean data, on ImageNet-100.

SRATE models offering higher robustness than corresponding RATE models and adv. training with SEA and SRATE offering much higher robustness under SEA attack and comparable robustness against other attacks.

| T=4 | CIFAR-10, Rate Encoding | | | | | CIFAR-10, Signed Rate Encoding | | | | |
|---|---|---|---|---|---|---|---|---|---|---|
| ATTACK | CLEAN | GN | FGSM | PGD | SEA | CLEAN | GN | FGSM | PGD | SEA |
| clean | 78.79 | 78.87 | 75.54 | 77.07 | 75.3 | **82.05** | 82.01 | 78.51 | 79.78 | 79.06 |
| gn | 79.23 | 78.58 | 75.92 | 77.51 | 75.12 | **82.13** | 81.83 | 78.26 | 79.7 | 78.01 |
| fgsm | 44.83 | 44.35 | 53.93 | 53.16 | 42.31 | 44.3 | 43.59 | 55.1 | 53.81 | **56.58** |
| pgd | 38.87 | 38.41 | 49.68 | 48.83 | 37.19 | 36.88 | 37.03 | 50.21 | 48.47 | **51.87** |
| sea, k=10 | 44.31 | 43.89 | 47.37 | 46.94 | 55.33 | 39.61 | 39.94 | 49.05 | 48.07 | **80.18** |
| sea, k=20 | 28.74 | 28.15 | 34.67 | 34.52 | 43.16 | 29.06 | 29.23 | 40.36 | 38.75 | **75.77** |
| Avg | 52.46 | 52.04 | 56.19 | 56.34 | 54.74 | 52.34 | 52.27 | 58.58 | 58.10 | **70.25** |
| T=4 | SVHN, Rate Encoding | | | | | SVHN, Signed Rate Encoding | | | | |
| clean | 85.66 | 85.84 | 85.87 | 86.04 | 85.01 | 89.44 | 89.59 | 89.64 | **89.87** | 83.98 |
| gn | 85.44 | 85.73 | 85.97 | 85.65 | 85.01 | 89.14 | 89.38 | 89.44 | **89.53** | 81.93 |
| fgsm | 44.09 | 44.15 | 50.26 | 48.33 | 41.94 | 43.37 | 44.06 | 49.94 | 48.23 | **67.57** |
| pgd | 38.82 | 38.77 | 45.03 | 43.31 | 35.82 | 35.46 | 35.43 | 42.45 | 40.53 | **65.76** |
| sea, k=10 | 37.81 | 37.61 | 38.13 | 37.89 | 54.06 | 34.62 | 35.10 | 35.26 | 35.13 | **92.27** |
| sea, k=20 | 21.46 | 21.75 | 22.36 | 22.63 | 32.66 | 23.55 | 23.72 | 24.93 | 24.97 | **87.21** |
| Avg | 52.21 | 52.31 | 54.60 | 53.98 | 55.75 | 52.60 | 52.88 | 55.28 | 54.71 | **79.79** |
| T=4 | CIFAR-100, Rate Encoding | | | | | CIFAR-100, Signed Rate Encoding | | | | |
| clean | 50.56 | 50.75 | 46.58 | 47.07 | 46.67 | **54.74** | 54.51 | 49.59 | 51.74 | 50.35 |
| gn | 50.03 | 51 | 46.41 | 46.72 | 46.15 | **54.6** | 54.06 | 49.64 | 51.93 | 49.69 |
| fgsm | 22.21 | 21.92 | 28.45 | 30.27* | 20.53 | 22.33 | 22.11 | 30.81 | 29.7 | **31.38** |
| pgd | 18.53 | 18.6 | 26.08 | 27.79* | 17.99 | 18.44 | 18.78 | **28.22** | 27.04 | 27.08 |
| sea, k=10 | 25.84 | 26.47 | 26.91 | 30.28 | 30.58 | 21.4 | 21.81 | 28.04 | 27.85 | **58.53** |
| sea, k=20 | 16.39 | 16.2 | 18.66 | 22.37 | 22.03 | 12.87 | 13.56 | 20.63 | 20.49 | **54.66** |
| Avg | 30.59 | 30.82 | 32.18 | 34.08 | 30.66 | 30.73 | 30.81 | 34.49 | 34.79 | **45.28** |

Table 6: A comparison of adversarially trained models (in columns) and attacks (in rows), with 'CLEAN' and 'clean' respectively denoting standard training and standard accuracy. The 'Avg.' row shows the average of accuracies in that column. A column-wise comparison between RATE and sRATE reveals improvement in standard accuracy and occasional compromise in robust accuracy, anticipated due to reduction in randomness. However, adv. training with SEA attack under sRATE, offers superior robust accuracy across all attacks. [ * obtained by BPTR, see Tab. 11 for BPTT vs. BPTR.]

## 6 CONCLUSION AND LIMITATION

Rate encoding is a standard communication model of neurons that was discovered very early in neuroscience (Gerstner et al., 2014). In computational models of spiking neurons, it plays an important due to their biological plausibility and robustness properties. In the present work, we explore novel aspects of rate encoding by quantifying their randomness and consequently trying to reduce it, which leads to the signed rate coding. The signed rate encoding has a biological plausibility due to its similarity with the excitatory and inhibitory neurotransmitters that help generate and suppress spikes. The proposed technique offers better convergence in training and improves clean accuracies. Further, the proposed computationally efficient and theoretically supported sparse adversarial attack in the encoding space shows the vulnerability of such a network if the attacker has access to the encoded input. Finally, adversarial training with encoding attack combined with signed rate encoding offers superior adversarial robustness, demonstrated by thorough empirical findings across three standard datasets. Despite its unique adversarial robustness properties and the proposed improvement in clean accuracy, there remains scope for further improvement in achieving state-of-the-art clean accuracy using rate-encoding models.

## ACKNOWLEDGEMENT

This work is part of the research project "ENERGY-BASED PROBING FOR SPIKING NEURAL NETWORKS" performed at Mohamed bin Zayed University of Artificial Intelligence (MBZUAI), in collaboration with Technology Innovation Institute (TII) (Contract No. TII/ARRC/2073/2021).

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

## A APPENDIX

### A.1 PROOF OF LEMMAS

**Lemma 1.** *The solution $\hat{\boldsymbol{\delta}}$ defined as per eqn. 16 is an optimal solution to the optimization problem in eqn. 15.*

*Proof.* We will give a proof by contradiction. Let us assume, that given any $\mathbf{z} \in \{0,1\}^{d \times T}$ there exists, $\tilde{\boldsymbol{\delta}}$ such that $\mathbf{z} + \tilde{\boldsymbol{\delta}} \in \{0,1\}^{d \times T}$ with $\langle \tilde{\boldsymbol{\delta}}, \mathbf{g} \rangle > \langle \hat{\boldsymbol{\delta}}, \mathbf{g} \rangle$. Now,

$$\langle \tilde{\boldsymbol{\delta}}, \mathbf{g} \rangle > \langle \hat{\boldsymbol{\delta}}, \mathbf{g} \rangle \iff \sum_{t=1}^{T} \sum_{i=1}^{d} (\tilde{\delta}_i^{(t)} - \hat{\delta}_i^{(t)}) g_i^{(t)} > 0$$

$$\implies \exists (t,i) : [[g_i^{(t)} > 0, \tilde{\delta}_i^{(t)} > \hat{\delta}_i^{(t)}] \text{ or } [g_i^{(t)} < 0, \tilde{\delta}_i^{(t)} < \hat{\delta}_i^{(t)}]]$$

In the rest of the proof, we omit the notation for time-step t, assuming it is understood.

If $g_i > 0, \tilde{\delta}_i > \hat{\delta}_i$, we have,

$$z_i = 0 \implies \hat{\delta}_i = 1 \implies \tilde{\delta}_i > 1 \implies \tilde{\delta}_i + z_i > 1 \implies \tilde{\delta}_i + z_i \notin \{0,1\}$$

$$z_i = 1 \implies \hat{\delta}_i = 0 \implies \tilde{\delta}_i > 0 \implies \tilde{\delta}_i + z_i > 1 \implies \tilde{\delta}_i + z_i \notin \{0,1\}$$

If $g_i < 0, \tilde{\delta}_i < \hat{\delta}_i$, we have,

$$z_i = 0 \implies \hat{\delta}_i = 0 \implies \tilde{\delta}_i < 0 \implies \tilde{\delta}_i + z_i < 0 \implies \tilde{\delta}_i + z_i \notin \{0,1\}$$

$$z_i = 1 \implies \hat{\delta}_i = -1 \implies \tilde{\delta}_i < -1 \implies \tilde{\delta}_i + z_i < 0 \implies \tilde{\delta}_i + z_i \notin \{0,1\}$$

As under all cases, we reach a contradiction and prove that no such $\tilde{\delta}$ can exist. $\qquad\square$

**Lemma 2.** *$\boldsymbol{\delta}^*$ obtained as per eqn. 17 is a maximizer of the optimization problem of eqn. 14.*

*Proof.* From Lemma 1 we know, that for a given $g_i^{(t)}$, there exists no $\tilde{\delta}_i^{(t)}$, such that, $\tilde{\delta}_i^{(t)} g_i^{(t)} > \hat{\delta}_i^{(t)} g_i^{(t)}$. Thus, for each coordinate $i$, we know that, $\hat{\delta}_i^{(t)}$ is an optimal choice. Let's assume there is another $k$-sparse vector, chosen from $\hat{\boldsymbol{\delta}}^{(t)}$ according to $\tilde{\pi}$ such that,

$$\sum_{i=1}^{k} \hat{\delta}_{\tilde{\pi}_i}^{(t)} g_{\tilde{\pi}_i^{(t)}} > \sum_{i=1}^{k} \hat{\delta}_{\pi_i}^{(t)} g_{\pi_i}^{(t)}$$

However, this is impossible as $\pi$ represents the sorting, $\hat{\delta}_{\pi_1}^{(t)} g_{\pi_1}^{(t)} \geq \hat{\delta}_{\pi_2}^{(t)} g_{\pi_2}^{(t)} \geq \cdots \geq \hat{\delta}_{\pi_d}^{(t)} g_{\pi_d}^{(t)}$. Thus, $\boldsymbol{\delta}^{*(t)}$ given by eqn. 17 is indeed an optimal solution. As the argument holds for any time-step $t$, $\boldsymbol{\delta}^*$ is an optimal solution. $\qquad\square$

### A.2 COMPARISON OF BUDGET FOR SIGNED RATE ENCODING

$$\mathbb{E}_{\mathbf{z} \sim \text{sBer}(\mathbf{x}), \hat{\mathbf{z}} \sim \text{sBer}(\hat{\mathbf{x}})} [\|\mathbf{z} - \hat{\mathbf{z}}\|_0]$$

$$= \sum_{i=1}^{d} \mathbb{E}_{z_i, \hat{z}_i} [\|z_i - \hat{z}_i\|_0] = \sum_{i=1}^{d} \mathbb{P}(z_i \neq \hat{z}_i) = \sum_{i=1}^{d} (1 - \mathbb{P}(z_i = \hat{z}_i))$$

$$= \sum_{i=1}^{d} (1 - \mathbb{P}(z_i = 1, \hat{z}_i = 1) - \mathbb{P}(z_i = -1, \hat{z}_i = -1) - \mathbb{P}(z_i = 0, \hat{z}_i = 0))$$

$$= \sum_{i=1}^{d} (1 - x_i^+ (x_i + \delta_i)^+ - x_i^- (x_i + \delta_i)^- - (1 - |x_i|)(1 - |x_i + \delta_i|))$$

$$= \sum_{i=1}^{d} (-x_i^+ (x_i + \delta_i)^+ - x_i^- (x_i + \delta_i)^- + |x_i| + (1 - |x_i|)|x_i + \delta_i|)$$

$$= \|\mathbf{x}\|_1 + \langle 1 - |\mathbf{x}|, |\mathbf{x} + \boldsymbol{\delta}| \rangle - \langle \mathbf{x}^+, (\mathbf{x} + \boldsymbol{\delta})^+ \rangle - \langle \mathbf{x}^-, (\mathbf{x} + \boldsymbol{\delta})^- \rangle \qquad (22)$$

### A.3 Additional Results

To further strengthen the study, we include results for the aggregated model (Sitawarin et al., 2022; Mukhoty et al., 2024). Let $\mathbf{z}_i = [\mathbf{z}_i^{(1)}, \mathbf{z}_i^{(2)}, \cdots, \mathbf{z}_i^{(T)}]$, where , s.t. $\mathbf{z}_i^{(t)} \sim Ber(\mathbf{x})$ with $\mathbf{z}_i^{(t)} \in \{0,1\}^d$. We report the test results using, $\frac{1}{m} \sum_{i=1}^{m} h(\mathbf{z}_i)$, with $m = 10$, approximating the smooth classifier, $\mathbb{E}_{\mathbf{z} \sim Ber(\mathbf{x})}[h(\mathbf{z})]$. The existing results reported earlier stands for $m = 1$.

The adversarial attacks are performed using the loss function, over the aggregated classifier. For fgsm and pgd attack, now there is a additional aggregation over $m$, that follows the back-propagation pipeline. While for SEA, the encoding space becomes $\{0,1\}^{d \times T \times m}$, where we have $\mathbf{z}_i \in \{0,1\}^{d \times T}$, $\boldsymbol{\delta}_i \in \{-1, 0, 1\}^{d \times T}$, and similar sparsity restrictions $\|\boldsymbol{\delta}_i^{(t)}\| \leq k$.

Results reported in Table 8 shows improvement clean accuracies when compared to 6 (uses $m = 1$), and attacks becoming stronger reducing robust accuracies in some cases. Despite aggregation improving attack strengths, SEA with sRATE offers superior robustness.

| T=4, m=10 | CIFAR-10, Rate Encoding | | | | | CIFAR-10, Signed Rate Encoding | | | | |
|---|---|---|---|---|---|---|---|---|---|---|
| ATTACK | CLEAN | GN | FGSM | PGD | SEA | CLEAN | GN | FGSM | PGD | SEA |
| clean | 84.1 | 83.4 | 79.36 | 81.6 | 79.02 | **85.51** | 85.3 | 81.37 | 83.41 | 82.7 |
| gn | 83.61 | 83.38 | 79.4 | 81.43 | 79.15 | **85.25** | 85.2 | 81.41 | 83.38 | 82.62 |
| fgsm | 40.06 | 39.84 | 49.19 | 50.39 | 40.89 | 40.43 | 40.43 | 51.76 | 49.37 | **51.91** |
| pgd | 33.27 | 32.88 | 43.96 | 43.72 | 34.03 | 32.96 | 33.12 | 45.17 | 43.3 | **45.17** |
| sea, k=10 | 50.47 | 49.79 | 51.46 | 53.29 | 60.18 | 45.63 | 45.69 | 55.14 | 53.51 | **85.67** |
| sea, k=20 | 35.97 | 35.86 | 40.16 | 41.22 | 49.06 | 35.83 | 35.59 | 46.89 | 44.95 | **81.77** |
| Avg | 54.58 | 54.19 | 57.26 | 58.61 | 57.06 | 54.27 | 54.22 | 60.29 | 59.65 | **71.64** |
| T=4, m=10 | SVHN, Rate Encoding | | | | | SVHN, Signed Rate Encoding | | | | |
| clean | 91.98 | 92.07 | 91.84 | 92.12 | 90.60 | 93.41 | **93.58** | 93.40 | 93.42 | 90.89 |
| gn | 91.93 | 91.998 | 91.77 | 91.89 | 90.63 | 93.32 | 93.42 | 93.33 | **93.43** | 89.94 |
| fgsm | 44.58 | 44.9 | 50.29 | 48.11 | 41.81 | 44.00 | 43.87 | 49.85 | 48.16 | **71.72** |
| pgd | 34.96 | 34.58 | 40.60 | 38.25 | 31.86 | 33.20 | 32.65 | 39.37 | 37.42 | **66.04** |
| sea, k=10 | 49.2 | 49.29 | 49.80 | 49.34 | 63.76 | 44.54 | 45.22 | 44.61 | 45.29 | **96.21** |
| sea, k=20 | 31.58 | 31.68 | 31.52 | 31.28 | 40.05 | 33.29 | 33.88 | 32.22 | 33.60 | **93.14** |
| Avg | 57.37 | 57.42 | 59.30 | 58.50 | 59.79 | 56.96 | 57.10 | 58.80 | 58.55 | **84.66** |
| T=4, m=10 | CIFAR-100, Rate Encoding | | | | | CIFAR-100, Signed Rate Encoding | | | | |
| clean | 55.09 | 55.33 | 50.15 | 50.2 | 49.93 | **58.17** | 58.53 | 53.19 | 55.18 | 55.16 |
| gn | 55.08 | 55.01 | 50.66 | 50.22 | 49.51 | **58.29** | 58.39 | 53.37 | 54.8 | 54.7 |
| fgsm | 26.47 | 26.69 | 29.83 | **33.35*** | 25.3 | 26.61 | 26.82 | 32.37 | 33.23 | 31.78 |
| pgd | 18.47 | 18.66 | 24.79 | **28.09*** | 18.66 | 18.55 | 18.79 | 27.55 | 27.03 | 25.16 |
| sea, k=10 | 34.4 | 34.76 | 32.25 | 34.63 | 33.91 | 31.96 | 31.76 | 34.19 | 35.15 | **64.73** |
| sea, k=20 | 27.2 | 27.35 | 27.51 | 29.85 | 28.9 | 24.52 | 24.65 | 30.01 | 30.45 | **61.45** |
| Avg | 36.12 | 36.30 | 35.87 | 39.16 | 34.37 | 36.35 | 36.49 | 38.45 | 39.31 | **48.83** |

Table 8: Inference performed using aggregation of model output with respect to input noise with $m = 10$, the attacks are updated accordingly. Similar to Tab.6, the results show improvement offered by sRATE, along with effectiveness SEA method in adversarial training.
* obtained by BPTR, as the it is lower than BPTT.

| Dataset | Encoding | Input | L1 | L2 | L3 | L4 | L5 | L6 | L7 | L8 | L9 | L10 |
|---|---|---|---|---|---|---|---|---|---|---|---|---|
| CIFAR-10 | RATE | 0.480 | 0.116 | 0.075 | 0.026 | 0.023 | 0.015 | 0.009 | 0.013 | 0.025 | 0.036 | 0.023 |
| | sRATE | 0.220 | 0.155 | 0.089 | 0.026 | 0.026 | 0.016 | 0.008 | 0.013 | 0.028 | 0.041 | 0.025 |
| CIFAR-100 | RATE | 0.480 | 0.151 | 0.139 | 0.049 | 0.051 | 0.035 | 0.021 | 0.036 | 0.059 | 0.043 | 0.066 |
| | sRATE | 0.230 | 0.167 | 0.161 | 0.048 | 0.049 | 0.033 | 0.020 | 0.035 | 0.060 | 0.047 | 0.070 |
| SVHN | RATE | 0.460 | 0.163 | 0.054 | 0.021 | 0.018 | 0.009 | 0.003 | 0.007 | 0.025 | 0.038 | 0.027 |
| | sRATE | 0.200 | 0.184 | 0.070 | 0.021 | 0.018 | 0.012 | 0.004 | 0.009 | 0.028 | 0.042 | 0.031 |

Table 9: Table reports the layer-wise spiking rate of the neurons for different datasets on VGGSNN architecture. It can be observed that sRATE offers lower average sparsity.

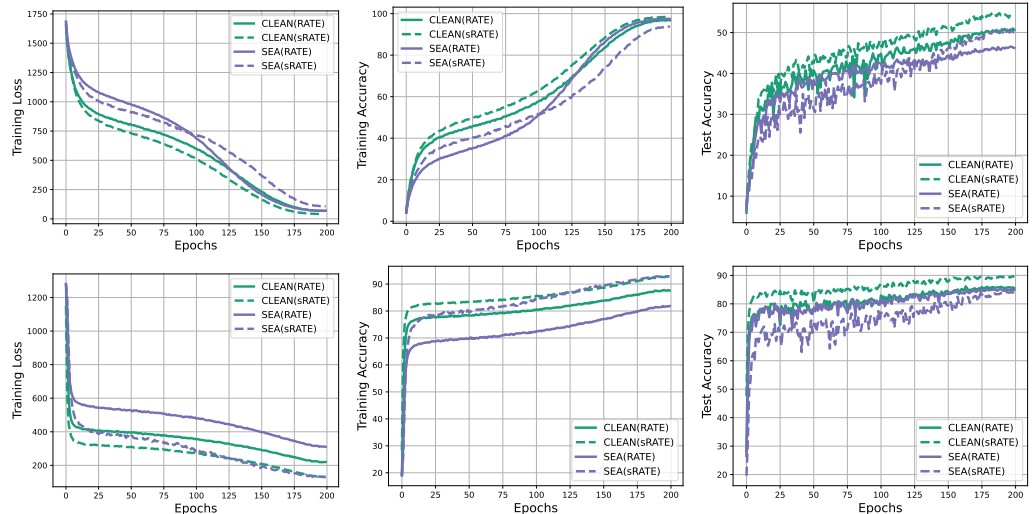

Figure 5: Comparison between rate encoding and signed rate encoding over training epochs reporting training loss, training accuracy and test accuracy for CIFAR-100 (top) and SVHN (bottom).

## A.4 TRAINING DETAILS

We list the specific hyper-parameters used to train our models in Tab. 10. For adversarial training with SEA on SVHN dataset, we use $k = 5$ for sRATE, but $k = 2$ for RATE, as RATE does not converge on $k = 5$.

Table 10: Training and testing hyper-parameters

|  | CIFAR-10/100 | ImageNet-100 | SVHN |
|---|---|---|---|
| Number epochs | 200 | 200 | 200 |
| Mini batch size | 128 | 128 | 128 |
| T | 4,6,8,10* | 4 | 4 |
| LIF: $\beta$ | 0.5 | 0.5 | 0.5 |
| LIF: $u_0$ | 0 | 0 | 0 |
| LIF: $u_{th}$ | 1 | 1 | 1 |
| Learning Rate | 0.1 | 0.1 | 0.1 |
| GN/FGSM/PGD (train): $\epsilon$ | 8/255 | na | 2/255 |
| GN/FGSM/PGD (test): $\epsilon$ | 8/255 | 8/255 | 8/255 |
| SEA (train): $k$ | 5 | na | 2, 5 |
| SEA (test): $k$ | 10, 20 | 10, 20 | 10, 20 |
| PGD (train): $\eta$ | 2/255 | na | 1/255 |
| PGD (test): $\eta$ | 2/255 | 2/255 | 2/255 |
| PGD (train) Iteration | 4 | na | 2 |
| PGD (test) Iteration | 8 | 8 | 8 |

Optimizer: SGD with momentum: 0.9, weight decay: $5 \times 10^{-4}$, rate Scheduler: cosine annealing

## A.5 SOFTWARE

The following is a list of the main libraries used in this work:

- Numpy (Harris et al., 2020)
- Pytorch (Paszke et al., 2019)
- Matplotlib (Hunter, 2007)
- Pandas (Reback et al., 2020)

We thank the creators and contributors of these open-source libraries for their contributions.

| T=4 | CIFAR-10, Rate Encoding (RATE) | | | | | CIFAR-10, Signed Rate Encoding (sRATE) | | | | |
|---|---|---|---|---|---|---|---|---|---|---|
| ATTACK | CLEAN | GN | FGSM | PGD | SEA | CLEAN | GN | FGSM | PGD | **SEA** |
| fgsm-bptt | 44.83 | 44.35 | 53.93 | 53.16 | 42.31 | 44.3 | 43.59 | 55.1 | 53.81 | 56.58 |
| fgsm-bptr | 58.52 | 57.69 | 66.8 | 63.45 | 59.97 | 55.3 | 54.89 | 61.95 | 60.87 | 64.28 |
| pgd-bptt | 38.87 | 38.41 | 49.68 | 48.83 | 37.19 | 36.88 | 37.03 | 50.21 | 48.47 | 51.87 |
| pgd-bptr | 54.89 | 53.69 | 65.15 | 60.46 | 57.05 | 48.97 | 50.04 | 58.73 | 57.39 | 60.51 |
| T=4 | SVHN, Rate Encoding (RATE) | | | | | SVHN, Signed Rate Encoding (sRATE) | | | | |
| fgsm-bptt | 44.09 | 44.15 | 50.26 | 48.33 | 41.94 | 43.37 | 44.06 | 49.94 | 48.23 | 67.57 |
| fgsm-bptr | 61.37 | 59.01 | 64.11 | 62.57 | 59.11 | 60.31 | 59.58 | 67.01 | 65.55 | 72.47 |
| pgd-bptt | 38.82 | 38.77 | 45.03 | 43.31 | 35.82 | 35.46 | 35.43 | 42.45 | 40.53 | 65.76 |
| pgd-bptr | 60.61 | 57.66 | 62.79 | 61.56 | 57.06 | 55.58 | 54.99 | 64.92 | 62.31 | 71.76 |
| T=4 | CIFAR-100, Rate Encoding (RATE) | | | | | CIFAR-100, Signed Rate Encoding (sRATE) | | | | |
| fgsm-bptt | 22.21 | 21.92 | 28.45 | 37.88 | 20.53 | 22.33 | 22.11 | 30.81 | 29.7 | 31.38 |
| fgsm-bptr | 27.82 | 28.16 | 32.53 | **30.27** | 28.02 | 26.81 | 28.61 | 33.67 | 33.9 | 34.92 |
| pgd-bptt | 18.53 | 18.6 | 26.08 | 35.93 | 17.99 | 18.44 | 18.78 | 28.22 | 27.04 | 27.08 |
| pgd-bptr | 25.16 | 24.92 | 30.45 | **27.79** | 25.13 | 23.78 | 24.96 | 30.64 | 31.75 | 32 |

Table 11: Comparison between BPTT and BPTR attack shows BPTT offers stronger attack in all cases except for CIFAR-100, where in PGD - adversarial training, BPTR constructs a stronger attack than BPTT, highlighted in bold.

| | T=6 | | T=8 | | T=10 | |
| CLEAN | RATE | sRATE | RATE | sRATE | RATE | sRATE |
|---|---|---|---|---|---|---|
| clean | 82.22 | 84.41 | 83.41 | 85.41 | 84.21 | 86.34 |
| gn | 81.3 | 84.14 | 83.47 | 85.42 | 83.76 | 85.65 |
| fgsm | 42.87 | 42.3 | 41.37 | 41.58 | 39.89 | 39.81 |
| pgd | 35.42 | 33.46 | 32.94 | 32.38 | 30.73 | 30.2 |
| sea, k=10 | 46.28 | 42.38 | 48 | 43.06 | 47.54 | 44.05 |
| sea, k=20 | 31.61 | 30.79 | 33.02 | 31.39 | 33.31 | 32.97 |

Table 12: Improvement in standard accuracy offered by signed rate encoding across different latency, as observed in training CLEAN model on CIFAR-10 dataset. We train separate models to handle inference latencies, which can be replaced by a single model following Anumasa et al. (2024).

| k | CLEAN | FGSM | PGD | SEA | CLEAN | FGSM | PGD | SEA |
|---|-------|------|-----|-----|-------|------|-----|-----|
| 2 | 69.6 | 67.53 | 68.18 | 70.23 | 66.78 | 67.43 | 67.81 | **79.99** |
| 4 | 61.29 | 60.77 | 61.14 | 65.78 | 56.63 | 60.45 | 60.27 | **81.03** |
| 6 | 54.89 | 55.55 | 55.51 | 61.75 | 49.23 | 55.92 | 55.32 | **81.21** |
| 8 | 48.64 | 50.54 | 51.01 | 58.19 | 44.08 | 52.61 | 51.44 | **80.97** |
| 16 | 33.54 | 38.39 | 38.51 | 47.75 | 32.3 | 42.9 | 41.56 | **77.6** |
| 32 | 19.88 | 26.17 | 26.27 | 33.75 | 22.99 | 35.61 | 33.91 | **69.31** |
| 64 | 11.64 | 16.93 | 17.98 | 20.74 | 17.03 | 29.58 | 27.83 | **53.26** |
| $\epsilon * 255$ | CLEAN | FGSM | PGD | SEA | CLEAN | FGSM | PGD | SEA |
| 2 | 71.67 | 71.19 | 72.15 | 68.16 | 73.86 | 73.09 | 74.44 | **74.51** |
| 4 | 63.27 | 65.53 | 66.29 | 60 | 64.41 | 67.78 | 67.95 | **68.72** |
| 6 | 53.66 | 59.42 | 60.03 | 51.47 | 54.09 | 60.99 | 60.41 | **62.23** |
| 8 | 44.6 | 53.67 | 53.63 | 42.04 | 44.33 | 55.01 | 53.81 | **56.01** |
| 16 | 16.8 | 32.7 | 30.83 | 16.72 | 20.04 | **35.32** | 32.25 | 33.7 |
| 32 | 3.33 | 10.66 | 12.38 | 4.27 | 8.75 | **18.85** | 17.26 | 13.85 |
| 64 | 4.94 | 8.48 | 9.42 | 5.16 | 7.88 | 14.32 | **13.45** | 10.23 |
| $\epsilon * 255$ | CLEAN | FGSM | PGD | SEA | CLEAN | FGSM | PGD | SEA |
| 2 | 70.98 | 70.15 | 70.78 | 66.88 | 72.75 | 72.21 | **73.3** | 73.06 |
| 4 | 59.36 | 62.21 | 63.11 | 56.52 | 59.9 | 64.53 | 63.99 | **64.67** |
| 6 | 47.7 | 55.53 | 55.14 | 45.76 | 46.51 | 56.77 | **54.96** | 56.84 |
| 8 | 38.33 | 49.17 | 48.76 | 36.94 | 36.74 | **51.01** | 48.47 | 50.82 |
| 16 | 23.85 | 39.11 | 38.04 | 22.74 | 22.56 | 39.45 | 36.42 | **40.34** |
| 32 | 17.04 | 33.03 | 31.23 | 16.99 | 17.6 | **34.05** | 30.7 | 33.31 |
| 64 | 10.93 | 25.04 | 24.04 | 11.99 | 12.55 | **25.93** | 23.6 | 23.03 |

Table 13: Corresponding to figure 4(a), (b) and (c) comparing different models from Tab. 6 under SEA (top), FGSM (middle) and PGD (bottom) attacks for different attack radius. Adversarial training with SEA under SRATE is found to offer the highest robustness across different attacks.

| $\epsilon$ | $\frac{2}{255}$ | $\frac{4}{255}$ | $\frac{6}{255}$ | $\frac{8}{255}$ | $\frac{16}{255}$ | $\frac{32}{255}$ | $\frac{64}{255}$ |
|---|---|---|---|---|---|---|---|
| fgsm (RATE) | 2.33 | 4.75 | 7.24 | 9.8 | 20.62 | 45.69 | 111.59 |
| pgd (RATE) | 2.38 | 4.93 | 7.44 | 9.79 | 17.09 | 27.82 | 54.01 |
| fgsm (SRATE) | 0.72 | 1.54 | 2.58 | 3.95 | 13.6 | 52.67 | 195.98 |
| pgd (SRATE) | 0.71 | 1.49 | 2.27 | 3.04 | 6.19 | 19.39 | 70.72 |

Table 14: sea budget $k$ comparable to $\epsilon$ obtained by finding $\delta$ through fgsm and pgd attack performed on CLEAN (RATE) and CLEAN (SRATE) models, as reported in Fig.4(d).

