# OpenReview forum: "Improving Generalization and Robustness in SNNs Through Signed Rate Encoding and Sparse Encoding Attacks"
_ICLR.cc/2025/Conference — ICLR 2025 Poster_

### Official Review · Reviewer_rMY3 · 2024-10-17

**Soundness:** 2
**Presentation:** 2
**Contribution:** 2
**Rating:** 5
**Confidence:** 4

**Summary:**

This paper proposes a new encoding method for SNN input, called Signed Rate Encoding (sRATE), which is experimentally shown to reduce the randomness of encoded inputs through statistical analysis. Additionally, the paper introduces a Sparse Encoding Attack (SEA).

**Strengths:**

S1. The sRATE is newly proposed and the analysis from randomness reduction is good.

S2. Authors conduct extensive experiments in evaluating the difference between RATE and sRATE SNNs, including adversarial training with various attacks.

**Weaknesses:**

W1. The theoretical analysis needs improvement. From the current analysis, it is unclear why sRATE (Eq. 12) reduces randomness compared to RATE (Eq. 9). Both x and $|x-\mu|$ (I believe the $x$ in Eq. 12 is actually $x-\mu$, though this is not explicitly stated in Section 3.2) belongs to [0,1], meaning that the actual ranges of $k1$ and $k2$ are the same. It is the experimental results (Table 1) that demonstrate the advantages of sRATE, which weakens the motivation behind the sRATE design.

W2. The novelty of the proposed SEA method seems limited. If I understand correctly, the solution in Eq. (16) appears to be a straightforward application of the FGSM attack to RATE/sRATE encoded inputs. The idea of selecting $k$ elements based on the top $k$ gradients in the SEA method is fairly straightforward. I am also somewhat confused by the results—since the perturbations generated by SEA can be considered a subset of those generated by FGSM (with SEA essentially removing some of the perturbations from the FGSM attack), it is unclear why SEA outperforms FGSM, as shown in Table 5.

W3. The writing could be improved. In my view, Lemma 1 and Theorem 2 present concepts that are fairly straightforward, and it may not be necessary to dedicate much space to them in the main text. Other writing-related comments (which do not affect my overall rating) are summarized in the Questions section.

**Questions:**

My primary concerns are outlined in the Weakness section. The following points are related to writing improvements and do not affect my overall rating.

Q1. As mentioned in W1, I believe the $x$ in Section 3.2 should actually be $x-\mu$, since $x$ could otherwise be negative. However, the authors did not clarify this point.

Q2. It seems unnecessary to define 'the positive and negative part' of a real number (Lines 197-199). For the sake of simplicity and readability, I believe it would be clearer to directly use piecewise linear functions instead, as this would make the formulation easier to follow.

Q3. The use of the same notation $k$ for different concepts is somewhat confusing. For instance, $k$ is used to represent randomness in Section 3, but it also denotes the sparsity of perturbations in Section 4.2 and the experiments.

Q4. The legend in Figure 4 is unclear, as it is difficult to distingusih between the dashed and solid lines in the legend.

---

> ### Author Response · Authors · 2024-11-23
>
> We thank the reviewer for the detailed questions. We have added a paragraph in section 3.2 (highlighted in blue) addressing one of the issues raised by the reviewer. We attempt to provide further clarification below:
>
> **W1: Theoretical reduction in randomness:**
>
>  A: We note the reduction in randomness offered by SRATE is a data-dependent result; such a reduction may not always happen. Following the suggestion of reviewer 2rdR, we have computed the rate of spikes of the neurons at each layer, which is reported in Tab. 5 and Tab. 9. We can observe that rate of spikes in the input layer has significant reduction in sRATE compared to RATE, which is possible when $|x-\mu| < x $ for most $x$, which also implies a reduction in randomness as the function $x(1-x)$ is an increasing function in $[0,0.5]$.
>
> To characterize formally, we observe, $|x -\mu| < x \iff 0 < \mu < 2x$. As $\mu \geq 0$ is true for $x \in [0,1]$, the condition fails only when $\mu > 2x$ or, $x>0.5$, thus, $x \in [\frac{\mu}{2}, 0.5]$ guarantees randomness reduction. However, there can be a reduction even outside this region. We have included this clarification in the paper.
>
> **W2: Part 1: Why SEA is stronger than FGSM:**
>
> **A:** FGSM is performed in the input space $[0,1]^d$, while SEA is performed in the encoding space $\\{0,1\\}^{d \times T}$. If the model receives an encoded input that is already corrupted, which is the case for SEA, the model processes it directly without noise injection. In contrast, since FGSM is in the input space, the input is encoded using Bernoulli and then processed by the network. The gradient for FGSM attack also has to back-propagate through the non-differentiable stochastic node, which is performed using the straight-through estimator. It is established that the stochasticity introduced by rate encoding is a source of robustness, which reduces the effectiveness of FGSM attacks. Further, since any change in the input space must pass through the encoding space, perturbations made in the encoding space subsume perturbations made in the input space under a comparable budget. Section 4.3 compares the budget between encoding space and input space.
>
> **W2: Part 2: SEA is an application of FGSM:**
>
> **A:**  Since both FGSM and SEA are single-step gradient-based attacks, they have similarities. However, we would like to highlight the subtle differences:
>
> FGSM is an attack in the input space with $l_\infty$-norm restriction, while SEA is an attack in the encoding space with $l_0$-norm restriction. More importantly, FGSM is not domain aware; thus, perturbed input $ x+\delta \notin [0,1]^d$ is required to be further project back to the input space. It uses two sequential projections, one for $l_\infty$, followed by projection to input domain, which makes FGSM sub-optimal, compared to a joint optimization that optimization both norm and domain restriction simultaneously (ref. [1]).
>
> In contrast, SEA optimizes domain and norm restrictions jointly. For example, in Table 2, if $sign(g_i)=1$, FGSM will suggest $\hat{\delta_i}=1$, irrespective of the value $z_i$. However, SEA being domain aware chooses $\hat{\delta_i}=1$ when $z_i=0$ and $\hat{\delta_i}=0$, when $z_i=1$, ensuring $\hat{\delta_i}+z_i \in \\{0,1\\}$. The case is more vivid in Table 3.
>
> The joint optimization of Eqn. (14) for SEA is thus different from FGSM optimization due to sparsity and domain constraints.
>
> **W3: A:** Following the suggestion from the reviewer, we have shifted the proofs to the appendix.
>
> **Q1: A:** Equation (12) is written in a general form without making mean-centering necessary, thus we mention $x_i \in [-1,1]$ at the start of section 3.2. To show a reduction in randomness after mean centering, we take the ratio between $k_1(x)$ and $k_2(x-\mu)$ and report it as $k_3(x)$, as given in Eq. (13). Table 1 shows the average values of $k_1(x), k_2(x-\mu),$ and $k_3(x)$. We highlight the changes made in section 3.2 in blue.
>
> **Q2: A:** The positive and negative part notation is required for deriving in Eq. (21) compactly. We carried the same notation everywhere.
>
> **Q3: A:** We use sub-script of $k_1, k_2, k_3$ for randomness, and $k$ for sparsity. It is especially so because, in section 4.3, we obtain sparsity k from the difference in randomness.
>
> [1] Francesco Croce and Matthias Hein. Mind the box: l1-apgd for sparse adversarial attacks on image
> classifiers. In International Conference on Machine Learning, pp. 2201–2211. PMLR, 2021.

---

> > ### Comment · Reviewer_rMY3 · 2024-11-25
> > **Comment after rebuttal**
> >
> > Thank you for the rebuttal. After reviewing the responses, I have decided to maintain my original score. The distinction between SEA and FGSM was clarified, which I found helpful. However, the rebuttal also confirms that the paper lacks theoretical proof for the claimed reduction in randomness, relying instead solely on data-dependent experimental evidence. This is a significant concern that affects my evaluation.

---

> > > ### Author Response · Authors · 2024-11-25
> > >
> > > The authors tend to disagree with the reviewer on the observation that there is no theoretical proof, while we give a characterization under what condition such a result will hold. Most proofs in machine learning are data-dependent; they hold only when the data follows a specific distribution/condition. In the present case, please observe in Table 1 that the mean of the pixels is around 0.4, so most of the pixels would follow the characterization $x \in [\frac{\mu}{2}, 0.5]$. Moreover, by computing $k_3(x)$, we can quantify how much randomness reduction a dataset will offer, even before training. Thus, it helps us choose the dataset where we would like to apply the method.
> > >
> > > It would be kind if the reviewer could re-evaluate the paper in this light.

---

### Official Review · Reviewer_iqc1 · 2024-10-31

**Soundness:** 3
**Presentation:** 2
**Contribution:** 3
**Rating:** 6
**Confidence:** 5

**Summary:**

The authors propose a signed rate encoding (SRATE) method that enables mean-centering of the input, reducing the randomness introduced by traditional rate encoding. This approach improves clean accuracy and enhances the generalization of rate-encoded spiking neural networks (SNNs).

**Strengths:**

The paper is well-structured and easy to follow.
The performance results show improvement with the proposed method.

**Weaknesses:**

Some details require further clarification.

**Questions:**

In the method section, is the encoding applied only to convert images to spike inputs, or does it also occur within spiking neurons in the network?

How could this technique be adapted for use with event-based datasets?

In Section 4.1, why does the attack rely on a first-order approximation rather than directly using the loss? What would happen if the direct loss were used instead?

In Figure 4 and Table 5, there are inconsistent notations (e.g., lowercase "fgsm" and uppercase "FGSM," similarly for "pgd" and "PGD"). Do these represent different methods, or is this just a formatting inconsistency?

How are FGSM and PGD applied to rate-based models in this study?

---

> ### Author Response · Authors · 2024-11-23
>
> We thank the reviewer for the reviewing effort and asking specific questions. We attempt to provide some clarification below:
>
> **Q1:** Is the encoding applied only to convert images to spike inputs, or does it also occur within spiking neurons in the network?
>
> **A:** The proposed encoding is only used to convert images to spikes. As described in Eq. (2), the neuronal activity remains the same.
>
> **Q2:** How could this technique be adapted for use with event-based datasets?
>
> **A:** The event-based data are sometimes collected into frames as a pre-processing step, where number of frames is the latency of the network.  In such a scenario, the SEA attack can be adopted owing to the discrete encoding space of the frames.
>
> **Q3:** Why does the attack rely on a first-order approximation rather than directly using the loss? What would happen if the direct loss were used instead?
>
> **A:** In the popular back-propagation-based training of neural networks, we evaluate the loss of the network on a specific input point and through back-propagation, we obtain a gradient of the loss at that point. Under such limitation, using the loss directly is equivalent to using a first-order approximation of the loss.
>
> **Q4:** In Figure 4 and Table 5, there are inconsistent notations (e.g., lowercase "fgsm" and uppercase "FGSM," similarly for "pgd" and "PGD"). Do these represent different methods, or is this just a formatting inconsistency?
>
> **A:** We agree that the naming convention looks confusing at first glance. However, we mention [in Experiments: Training] that "Since both attack and adversarially trained models have the same name, we denote the attacks with small cases, while the models are highlighted in the capital within the figures and Tab. 5."
>
> We considered appending "-AT" to the model names to highlight adversarial training and to distinguish it from the corresponding attack, but that increases the width of Tab. 5 beyond the page width.
>
> **Q5:** How are FGSM and PGD applied to rate-based models in this study?
>
> **A:** They are implemented using the Straight-Through-Estimator, which assumes an identity function to perform back-propagation through the non-differentiable stochastic node (mentioned under Experiments: Competitors). It enables us to compute the gradient of these attacks in the input space. Some additional details are provided in answering Q2 of the reviewer 2rdR.

---

> > ### Comment · Reviewer_iqc1 · 2024-11-26
> >
> > I think the author has addressed most of my concerns. Thank you.

---

> > > ### Author Response · Authors · 2024-11-27
> > >
> > > We highlight some additional results that were introduced in the process of rebuttal  (given in blue in the paper):
> > >
> > > i. Adversarial training with SEA with sRATE continues to outperform other methods even under improved adversarial attack that uses model aggregation (reported in Table 5 for m=1 and Table 8 for m=10),
> > > ii. sRATE offers higher clean accuracy under model aggregation (reported in Table 8)
> > > iii. sRATE produces significantly fewer spikes in the input layer, justifying a reduction in the probability of spikes due to mean centering (Table. 5) and comparable spiking at the network level.
> > >
> > > In light of these additional results and the clarification provided in the rebuttal process, would the reviewer consider re-evaluating the score for the paper?

---

### Official Review · Reviewer_2rdR · 2024-11-01

**Soundness:** 3
**Presentation:** 4
**Contribution:** 3
**Rating:** 8
**Confidence:** 5

**Summary:**

This paper aims to explore an input encoding that balances the robustness and generalization on clean input of the model. The authors propose the signed rate encoding (SRATE) method that improves the accuracy of the model. In addition, they also introduce the sparse encoding attack on the SRATE input. Through theoretical analysis and experiments on datasets like CIFAR-10 and CIFAR-100, the authors demonstrate that SRATE improves generalization, while adversarial training with SEA offers superior robustness compared to traditional methods.

**Strengths:**

The proposed method utilizes the rate-coding input, which is not only more bio-plausible but also provides inherent advantages in robustness and sparsity, distinguishing it from models that rely on constant input encoding methods.

The authors provide a solid theoretical basis for the SRATE, demonstrating that it preserves the essential characteristics of Poisson input encoding while effectively reducing randomness.

The authors proposed a novel attack method, offering a solution for finding optimal adversarial examples under binary and sparsity constraints.

**Weaknesses:**

One drawback of this method is its relatively low accuracy on clean inputs, with even the non-adversarially trained model achieving only ~55% accuracy on the CIFAR-100 dataset. The significant drop in clean accuracy may not justify the robustness gains, which raises the questions about the effectiveness of this approach, particularly when other methods (e.g. adversarial training) may offer a better balance between performance and robustness.

**Questions:**

Sparsity is a critical feature in neural systems, offering benefits in terms of computational efficiency and reduced energy consumption. Given that this model is based on rate encoding, I wonder whether the model with SRATE encoding also leverages these advantages?

It would be valuable to know if the authors considered using the Expectation Over Transformations (EOT) method for adversarial attacks, especially since EOT is an effective approach to deal with models that incorporate randomness.

---

> ### Author Response · Authors · 2024-11-23
>
> We are thankful to the reviewer for a detailed review of the paper and insightful comments; it helped us improve the paper. The additional results included in the paper are highlighted in the colour blue. We list our answers below:
>
> **Q1: Sparsity of spikes for RATE vs. sRATE:**
>  We compute the layer-wise sparsity of spikes in the neurons (number of times a neuron spikes, divided by latency T, and then averaged over the neurons in the layer) for both RATE and sRATE. Tables 5 and 9, following the reviewer's suggestion, report the statistics for different datasets and respective networks. We can observe from Table 5 that spikes in the input layer have reduced significantly in sRATE due to mean centering. Further, the spike rate averaged over all the neurons shows that sRATE raises the spiking rate only slightly.
>
> **Q2: EoT for adversarial attacks:** It was observed that certain transformations,  e.g., shifting of view-point, jpeg compression, noise injection, etc. when applied to the adversarial input image, can mitigate the effectiveness of the adversarial perturbation. The Barrage of Random Transformation (BaRT, ref. [3]) was proposed as a defense strategy against adversarial attacks, where randomly transformed input images are supplied to the same model in parallel, and the aggregated output is considered as the final output. However, the adversarial input in such cases was oblivious to the transformation to be applied. Consequently, under EoT, it was noted that a white-box adversarial attack that is aware of the possible set of random transformations can improve the adversarial attack by incorporating the random transformations in the back-propagation pipeline.
>
>  In the present case, the input image (possibly adversarial) is rate encoded through Bernoulli stochasticity, and the gradient in the input space (required to compute FGSM, PGD attack) is back-propagated through the non-differentiable stochastic node using the Stright-Through-Estimator (mentioned in Experiments: Competitors) and aggregated over $T$ steps to return to $d$ dimensional input space from $d \times T$ dimensional encoding space, following back-propagation. Thus, the FGSM and PGD attacks are in line with EoT in the sense that they are aware of the noisy transformation. Note that the SEA is performed in the encoding space, not requiring the gradient to be back-propagated to the input space.
>
> To further strengthen the study, Tab. 8, included in the appendix, reports results for the aggregated model (following Eq. (3) in ref. [2], Eq. (9) in ref. [3]). Using exiting notation, let $z_i = [ z_i^{(1)},  z_i^{(2)}, \cdots , z_i^{(T)}]$, s.t. $ z_i^{(t)} \sim Ber( x)$ with $ z_i^{(t)} \in \\{0,1\\}^d$. We report the test results using the average over the model output, $\frac{1}{m}\sum_{i=1}^m h_{\theta}( z_i)$, with $m=10$, which approximates the smooth classifier, $E_{{z \sim Ber(x)}}[h_{\theta}(z)] $, while the other existing results in the paper stands for $m=1$.
>
> The adversarial attacks are performed using the loss, $\mathcal{L}(\frac{1}{m}\sum_{i=1}^m h(z_i), y)$. For FGSM and PGD attacks, now there is an additional aggregation over $m$ that follows the back-propagation pipeline. While for SEA, the encoding space becomes $\\{0,1\\}^{d \times T \times m }$. Following line 263, for SEA we have $z_i \in \\{0,1\\}^{d\times T}$, $\delta_i \in  \\{-1, 0,1\\}^{d\times T}$, and similar sparsity restrictions $||\delta_i^{(t)}||_0\leq k$. Tab. 8 shows (i) improvement in clean accuracies when compared to $m=1$ and (ii) clean accuracy of sRATE being higher compared to RATE. (iii) Despite improvement in attack strengths, SEA (sRATE) offers superior robustness in most cases.
>
> **W1: Clean accuracy of rate-encoding:** We partially agree with the reviewer's observation. However, improving clean accuracy offered by rate-encoding is one of the objectives of the proposed method, and adversarial training has the disadvantage of being computationally expensive for large datasets. As reported in Tab. 8, using a smooth classifier in inference can further push the clean accuracy for all the datasets.
>
> --------------
> [1] Raff, Edward, et al. "Barrage of random transforms for adversarially robust defense." Proceedings of the IEEE/CVF Conference on Computer Vision and Pattern Recognition. 2019.
>
> [2] Sitawarin, Chawin, Zachary J. Golan-Strieb, and David Wagner. "Demystifying the adversarial robustness of random transformation defenses." International Conference on Machine Learning. PMLR, 2022.
>
> [3] Mukhoty, Bhaskar, et al. "Certified Adversarial Robustness for Rate Encoded Spiking Neural Networks." The Twelfth International Conference on Learning Representations.

---

> > ### Comment · Reviewer_2rdR · 2024-11-25
> > **Reply**
> >
> > I believe the authors have addressed most of my concerns. It is good to see that the SRate method demonstrates an advantage in sparsity over conventional rate-coding approaches. The explaination about SEA that already incorporates EoT is also convincing. It will be beneficial if the authors discuss more about the "clean accuracy issue" and highlighting potential future directions to address this problem in the _Conclusion and Limitation_ section of the manuscript.

---

> ### Author Response · Authors · 2024-11-25
>
> We thank the reviewer for the kind re-evaluation of the paper. We have updated the *Conclusion and Limitation* section per the suggestion.

---

### Meta-Review · Area_Chair_8ffW · 2024-12-20

**Metareview:**

The authors propose a signed rate encoding (SRATE) method that enables mean-centering of the input, reducing the randomness introduced by traditional rate encoding. The reviewers acknowledged a solid theoretical basis for the SRATE as well as extensive experiments in evaluation. One reviewer pointed out the lack of proof, but the main problem was actually about some clarifications in the current proof, which was raised by another reviewer and then addressed by the author in the rebuttal. Thus, I recommended an acceptance.

**Additional Comments On Reviewer Discussion:**

The reviewers gave positive feedback on the authors' rebuttal.

---

### Decision · Program_Chairs · 2025-01-22

Accept (Poster)